# Development and Validation of the Adolescent Sexual and Reproductive Competency Assessment Tool (ASRH-CAT) for Healthcare Providers

**DOI:** 10.3390/healthcare11081116

**Published:** 2023-04-13

**Authors:** Intan Kartina Abdul Karim, Rosnah Sutan, Azmi Mohd Tamil, Norizan Ahmad

**Affiliations:** 1Community Health Department, Medical Faculty, University Kebangsaan Malaysia, Bandar Tun Razak, Cheras, Kuala Lumpur 56000, Malaysia; 2Kedah State Health Department, Ministry of Health, 1358, Jalan Kuala Kedah, Taman Teratai Jingga, Alor Setar 05400, Kedah, Malaysia

**Keywords:** sexual health, reproductive health, primary care, competency skills, healthcare, public health, validation, adolescent

## Abstract

Objectives: This study aimed to validate a competency assessment tool for adolescent sexual and reproductive health (ASRH) services for healthcare providers (HCP) at primary healthcare (PHC) facilities that require a specific set of competency skills to address ASRH problems. Methods: The tool development process used the nine steps of scale development and validation. Fifty-four items were yielded through the expert panel discussion. Two hundred and forty respondents were recruited for an online questionnaire using non-probability sampling. The item content validity index (I-CVI) and exploratory factor analysis (EFA) were used for construct validity. Results: Fourteen items were removed based on the I-CVI (scores < 0.8) and two items were removed in the EFA (factor loadings < 0.4). The reliability analysis, according to the latent factor, yielded a good item-total correlation (ITC) and a good internal consistency value, with Cronbach’s alpha values of 0.905–0.949. Conclusions: The final ASRH competency assessment tool (ASRH_CAT) contains 40 items and is reliable and suitable for use in studies related to the ASRH competency assessment of HCPs at the PHC level.

## 1. Introduction

Globally, sexual and reproductive health (SRH) is a leading public health problem among the early-reproductive-age group, which requires appropriate services for preventive measures [1,2]. Adolescent SRH (ASRH) problems are associated with a lack of SRH knowledge; socio-cultural norms regarding sexual activity; healthcare workers’ negative behaviors; and the awareness, availability, cost, and quality of the SRH services provided [1,2,3,4,5,6]. Women face higher SRH risk, as their health status determines their potential offspring, but there is scarce evidence that they have been targeted in the literature, especially among young adults [1,3]. Unmet SRH needs are always emphasized in adolescent health intervention, but the literature measuring healthcare providers’ (HCP) competency skills in delivering SRH services is scarce [6,7,8]. Little is known about healthcare workers’ management of personalized care for adolescents regarding SRH and their competency regarding risk screening, decision-making, and creating a shared care plan. The Countdown 2030: Drivers Technical Working Group highlighted that SRH is the key aspect of the disease burden of adolescents, for which challenges should be overcome [8]. The provision of adolescent-friendly health services by HCP is listed as a core health sector mandate that requires service delivery monitoring. The ability of HCP to advocate healthy lifestyles to promote good SRH screening and identify those in need of treatment and follow-up is crucial. Investing in dedicated, competent staff will foster the management of adolescent SRH and the readiness to initiate programs with multisectoral collaboration to advocate good SRH. 

Malaysia is a transitional, middle-income country with a large population of younger people. Adolescents (aged 10–19 years) comprise approximately 17% (5.5 million) of the Malaysian population and are known as the healthiest population but require education on risk prevention [9]. Malaysian primary health clinics (PHCs) have integrated adolescent health services for the past two decades. However, PHCs are poorly utilized for SRH problems among adolescents [7]. The Malaysian Population and Family Survey reported that more than half of the participating adolescents had adequate knowledge of reproductive organs. However, the majority lack knowledge of SRH risks and sexually transmitted infections (STIs) [7,9].

Local and global studies have demonstrated the association between poor adolescent SRH knowledge and practice with the incidence of STIs, unintended pregnancies, unsafe abortions, and baby abandonment [3,6,10]. Adolescents require motivated and trained HCPs to provide physiological, cognitive, emotional, and social support in facing the transition period into adulthood [8,11]. Nevertheless, many HCPs do not possess the appropriate level of knowledge, confidence, and skills [5,12]. Adolescents often find public healthcare services unacceptable due to the perceived neglect of privacy and confidentiality services. The reasons for not using public healthcare services are associated with the HCPs’ poor attitudes and lack of adolescent-friendly resources, such as educational materials and counseling [10,11,12].

Currently, Malaysian PHCs practice integrated health services to cater to multiple health issues in a diverse community with multifaceted adolescent problems [7]. Hence, strategies for establishing and strengthening the network and effective communication between agencies and health services have been implemented for the holistic intervention of adolescent health problems. HCPs need to run maternal and child health programs as the priority preventive program. Therefore, a shortage of staff and resources may compromise their competency in focusing on ASRH services [12]. Hence, a coordinated and integrated multidisciplinary strategy for an adolescent health program must be established through a shared network to strengthen effective communication.

Well-trained, competent HCPs can implement the best possible practices in ASRH at their healthcare facilities. Public health evidence has placed less priority on emphasizing HCP competency skills in providing ASRH services [13,14,15]. Competent HCPs may help reduce barriers and encourage adolescents to seek medical attention for their health problems, especially regarding ASRH, which may be perceived as indecent and taboo in many cultures [10,14]. Addressing ASRH requires specific competency skills training that uses proper guidelines to assist HCPs in providing quality services [15]. ASRH-competent HCPs will meet caregiver needs by enhancing access to care and reducing health disparities among adolescents [16]. Formal competency skills training with certification during on-the-job training will ensure the quality of ASRH services [17].

Without appropriate practice, competency skills will lack direction. Appropriate assessments and monitoring require comprehensive content to measure the success and progress of staff in implementing ASRH services. Using self-assessment evaluation for assessing HCP competency will encourage staff to improve their performance. The World Health Organization (WHO) outlined the core competencies for SRH in primary care [18]. HCPs should attain four domains consisting of 13 main competencies [18,19]. It is crucial to have a specific decision-making and planning component assessment tool in the context of ASRH needs [19]. To the best of our knowledge, there is no ASRH competency assessment tool for assessing HCP decision-making ability at the PHC level. The HCP should be aware of the importance of connecting various stakeholders with a focus on existing adolescent health initiatives. It was suggested that reviewing and building capacity at individual, organizational, and system levels to strengthen the management of adolescent program initiatives, such as SRH programs, would be timely [8]. Subjective perceptions of life values across local cultural and ethnic beliefs will improve the target group’s acceptance and service usage and reduce service inequalities. A competent HCP will ensure the quality of care, establish a relationship with the adolescent, and promote open communication [9]. Concerns regarding the disclosure of diagnosis and treatment require the HCP to undergo a certification training program to address adolescent needs. The challenges include providing SRH and managing adolescent cases with an understanding of the local cultural and developmental stage of adulthood from the identity stage to the role of confusion and from the intimacy stage to isolation and full independence [9]. The National Research Council and Institute of Medicine report on adolescent health services (2009) stated that most HCPs working at PHC are nurse practitioners via adolescent health-friendly clinics and school health services led by family physicians or medical doctors. Therefore, the present study aimed to develop and validate an ASRH competency assessment tool (ASRH_CAT) tailored to the local socio-cultural needs and diverse HCP levels that serve at the PHC level.

## 2. Materials and Methods

### 2.1. Study Design and Settings

The present validation study was aimed at developing a novel ASRH competency questionnaire. The study was conducted over 12 weeks from June 2021 to August 2021, which coincided with the fourth wave of COVID-19 in Malaysia [20]. Therefore, respondents were invited by various Malaysian HCPs, who were attached to PHCs, to participate in an online survey. The population-based survey was carried out during the Movement Control Order (MCO) of the fourth wave of COVID-19 when Malaysia went into strict lockdown. Various COVID-19 pandemic management activities, including COVID-19 screening at the COVID-19 assessment centers (CAC), isolation and treatment at the COVID-19 quarantine and treatment centers (PKRC), and the national COVID-19 immunization program (PICK) were ongoing at the public PHCs and involved many HCPs. Therefore, all respondents were recruited using an online platform via email and WhatsApp blasts linked to a Google Form. The online platform was chosen to reduce the risk of in-person contact that would have been vulnerable to droplet transmission of the COVID-19 virus. The coverage of internet access and digital media among the HCPs, as well as many healthcare programs delivered during the pandemic, were planned using a digital appointment system. 

### 2.2. Questionnaire Development

The ASRH competency skills assessment tool (ASRH_CAT) was developed in this study as a new assessment tool for assessing basic HCP competency in decision-making and planning ASRH services at PHCs. A self-administered, online, bilingual (English and Malay) survey was conducted involving healthcare practitioners, which included doctors, nurses, and medical assistants working at public PHCs. The questionnaire development was divided into three phases adapted from the best practices for developing and validating scales for health, social, and behavioral research [21].Phase 1: Item developmentStep 1: Domain identification and item generation

We generated domains and identified items using literature evidence. A team of five experts (one public health specialist, two family health specialists, one obstetrics and gynecology specialist, and one adolescent psychiatrist) was appointed. All experts were contacted directly, and an in-depth interview using video calling was conducted with each expert using semi-structured questions to determine the domains for the new tool. Items were mainly generated using a deductive method focusing on the core competencies needed.

To construct the tool, the items were developed based on five guidelines regarding core competencies and decision-making for ASRH management in PHCs [13,17,19,22,23]. Later, the tool was screened for repetitiveness, complexity, and irrelevance before being included in the items pool. Four domains were determined to have the most agreement among all five experts: (1) ability to provide ASRH education (HE), (2) self-perceived capability (C), (3) self-perceived knowledge level (K), and (4) self-perceived attitude (A). The questionnaire was drafted in English and then translated into Malay using forward and backward methods [24,25]. The translation involved two linguists fluent in Malay and English.Step 2: Content validity

An evaluation, completed by the experts and target population, was performed. The experts evaluated each item constituting the domain for content relevance, representativeness, clarity, and consistency to determine the item content validity index (I-CVI). Fifteen people from a targeted population group participated in evaluating each item constituting the domain based on their life experience handling the adolescent group.

The initial draft included 54 items, and 14 items were removed based on the expert consensus agreement due to irrelevance, redundancy, and unclear statements. The remaining 40 items were scrutinized and rephrased to eliminate complexity, as suggested by the expert group. The final 40 items were able to measure the basic competency needed by HCP according to the Ministry of Health Malaysia guidelines [13,17,19,22,23] for implementing ASRH services at PHCs.Phase 2: Scale development (construct validity)Step 3: Pre-testing questions (cognitive interviews)

A total of 40 items were included in the items pool. All items were constructed in the form of statements and the respondents rated their self-perceived competency level using a 5-point Likert scale (0 = strongly not confident and 4 = strongly confident). Subjective responses during the answering process were reflected through their comments.Step 4: Survey administration and sample size

To estimate the sample size for the validation study, assumed indiscriminately, each item used was assessed by 5–10 people. The Kaiser–Meyer–Olkin (KMO) sampling adequacy test was used to ensure an adequate sample size [26,27]. Therefore, this study targeted a minimum of 200 samples, and we managed to recruit 240 respondents. A cross-sectional study was conducted for exploratory factor analysis using state healthcare workers serving PHC facilities under the Kedah State Department of the Ministry of Health of Malaysia. Kedah state is located in northern Malaysia and has rich socio-cultural practices that influence lifestyle behaviors in society.Step 5: Item reduction

The proportion of items with complete responses was determined. According to Moret et al. [27], the psychometric analysis of the scale can be optimized when items with many missing responses are deleted to ensure the availability of complete cases for scale development and improve the item response distribution by reducing the ceiling effect. This is the first questionnaire that measures healthcare workers’ perceptions of their competency in managing SRH. Therefore, we planned to conduct a subsequent study using a diverse population when the pilot study had been completed to assess the questionnaire’s validity. During scale development (content validity), we omitted items with responses that were not available or applicable.Step 6: Extraction of factors: Data

An exploratory factor analysis (EFA) was conducted to determine the optimal number of factors or domains that fit a set of items.Phase 3: Scale evaluationStep 7: Tests of dimensionality

We created scale scores to allow for reliability and validity analysis. The ASRH_CAT has 40 items and 4 domains with a range in scores of 0c100. The scale scores were calculated, and the mean of the raw item scores was computed. Each domain is calculated independently. A higher score indicates a better competency level. The calculation for each domain is as follows:Domain 1 (HE1–HE13): Self-perceived ability in providing ASRH education; sum of item scores in domain 1 × 100/(13 × 4)Domain 2 (C1–C11): Self-perceived capability in ASRH management: sum of item scores in domain 2 × 100/(11 × 4)Domain 3 (K1–K11): Self-perceived adequate ASRH knowledge in decision making: sum of item scores in domain 3 × 100/(11 × 4)Domain 4 (A1–A5): Self-perceived appropriate attitude in ASRH management: sum of item scores in domain 4 × 100/(5 × 4)

Aggregation scores for all domains are not calculated, as the competency level needs to be assessed based on the domains.Step 8: Reliability testing

We conducted test–retest reliability at 2-week intervals with the same set of respondents to establish whether the responses were consistent when repeated. The internal consistency of the scale was estimated with Cronbach’s alpha [28].Step 9: Test of validity

We assessed the feasibility (relevancy, representativeness, clarity) of the questionnaire to be used (face validity). We recruited six respondents (not involved in the real study) and used a scale of 1–4, where 1 = not relevant, 2 = somewhat relevant, 3 = quite relevant, and 4 = highly relevant for each item. We analyzed the test using Cohen’s kappa index for inter-rater agreement and used the results to guide questionnaire improvement.

We conducted a pre-test of 20 healthcare workers who were not involved in the recruitment of the true study. The validity assessment of the questionnaire was performed using the difficulty index and the item discrimination index. First, the validity of the questionnaire was assessed based on the proportion of respondents who answered the items correctly (item difficulty index). A higher value for the item difficulty index indicated that more respondents were practicing competently. The item discrimination index measured how well an item could differentiate between respondents who were competent vs. not competent in managing ASRH cases.

This set of questionnaires is the first competency tool available to assess the self-perceived competency level in healthcare providers’ management of ASRH care. Therefore, no published, validated study instrument can be used as a benchmark to assess concurrent validity for the same purpose.

### 2.3. Statistical Analysis

Statistical analysis was performed using SPSS v.26. All 40 items were computed in the analysis. Varimax oblique rotation was used for the EFA measurement. The number of factors retained was determined using the Kaiser criterion with eigenvalues > 1. Items were suppressed when the factor loading was <0.4. The Kaiser-Meyer-Olkin (KMO) index was used for sample adequacy, with a significant Bartlett’s test for sphericity valuation. A reliability analysis was performed, and Cronbach’s alpha was determined.

## 3. Results

### 3.1. Content Validity

The expert panel discussion yielded a total of 54 items, as presented in Table 1. The questionnaire contained four domains: (1) self-perceived ability in ASRH education (HE), (2) self-perceived capability in ASRH education (C), (3) self-perceived knowledge (K), and (4) self-perceived attitude (A), with 13, 20, 11, and 10 items, respectively. For the domain of self-perceived ability in ASRH education, the highest mean score was for item HE7 (3.76, SD 0.909). For the domain of self-perceived capability in ASRH education, items C9, C10, and C11 had the highest mean score of 3.70. For the domain self-perceived knowledge, item K7 had the highest mean score of 3.70 (SD 0.825). For the domain self-perceived attitude, item A5 yielded the highest mean score of 2.90 (SD 0.850).

Table 1 depicts the I-CVI scores for consistency, representativeness, relevance, and clarity. For consistency, only two items obtained a low score: C15 (0.60) and C18 (0.40). For relevance, five items recorded low scores: C11 (0.60), C14 (0.60), C15 (0.60), C16 (0.60), and A2 (0.60). For representativeness, six items recorded low scores: C6 (0.60), C11 (0.60), C16 (0.40), C19 (0.60), A5 (0.4), and A9 (0.40). For clarity, 11 items had low scores: C2 (0.40), C5 (0.40), C6 (0.40), C11 (0.60), C14 (0.60), C19 (0.40), A1 (0.40), A2 (0.40), A5 (0.40), A7 (0.40), and A9 (0.40). These items were removed due to low I-CVI scores (<0.80).

### 3.2. Construct Validity (EFA)

A cross-sectional study was conducted among the 240 healthcare workers recruited from Kedah State (Table 2). Most of the respondents were Malay women, and the mean age was 39.6 years (SD 7.1). More than half of the respondents possessed at least a diploma and worked as a nurse. The mean length of service in the public health sector was 12.46 years (SD 6.50).

A cross-sectional study was conducted for exploratory factor analysis. The KMO index for the questionnaire was 0.94, which indicated that the sample size employed in the validation study was sufficient to run the analysis. The *p*-value for Bartlett’s test of sphericity was significant at *p* < 0.001, signifying the presence of multidimensionality in all the items.

The EFA revealed that four latent factors were detected, as shown in Table 3. The reliability analysis, according to the latent factor, yielded a good item–total correlation (ITC) and a good internal consistency value, with Cronbach’s alpha values of 0.905–0.949. Only two items from the domain self-perceived ability to provide ASRH education (HE8 and HE11) were removed due to low factor loadings (<0.40).

### 3.3. Reliability Testing 

During the pre-test study, we conducted test–retest reliability at 2-week intervals that involved six healthcare workers who were involved in SRH services to establish whether responses were consistent when repeated. The internal consistency of the scale was estimated with Cronbach’s alpha. The results obtained ranged from 0.8 to 0.9 for each item.

### 3.4. Face Validity 

We assessed the feasibility of the questionnaire based on the items’ relevancy to the domains, the items’ representativeness of the study objectives, the items’ relevancy to the concepts related to the study topic, and the clarity of words or terms used (face validity). During the pre-test study, we recruited six healthcare workers who were involved in SRH services and used a scale of 1–4: 1 = not relevant, 2 = somewhat relevant, 3 = quite relevant, and 4 = highly relevant. We analyzed the results using Cohen’s kappa statistic for inter-rater agreement. Cohen’s Kappa for relevancy to domains was 0.9, representatives to objectives were 0.8, relevancy to study topic was 0.7, and clarity was 0.8. The format is user-friendly and takes less than 10 min to complete. It is self-explanatory and can be administered via an online survey or manually using paper. 

### 3.5. Item Difficulty Index 

We conducted a pre-test among 20 healthcare workers who were not involved in the recruitment for the true study. Item analysis was conducted for the question items in each domain. Subsequently, respondents with a score of 50% of correct items within the domain were assigned a 1 for a correct domain, while respondents with a score of <50% were assigned 0 (for an incorrect score for the domain). The item for the domain difficulty index was calculated and ranged from 0.4 to 0.7 (Table 4). Therefore, no domain was omitted from the questionnaire. No items were omitted from any of the domains, as the item difficulty index > 0.6. The item discrimination index was calculated and ranged from 0.4 to 0.6; therefore, the domain was considered appropriate to be used.

## 4. Discussion

The WHO created the core competencies manual to assist HCPs in guiding adolescents in receiving quality SRH services regardless of their background. However, due to sociocultural diversity, some countries may need to tailor the competency assessment according to their socio-cultural background, local policies, and HCP training availability. The Adolescent Sexual and Reproductive Competency Assessment Tool (ASRH-CAT) is a self-administered questionnaire for HCPs that was newly designed explicitly for ASRH services at Malaysian PHCs. The tool was developed based on WHO guidelines, and the services delivered at adolescent health clinics were tailored to local guidelines. Therefore, the tool might be applicable to be used by any country interested in assessing HCP competency levels before enrolling them in specific SRH courses or placing them in the ASRH services to cater to adolescent health. The ASRH-CAT can be used to assess HCP readiness to be in charge of the adolescent health program. The questionnaire items were composed based on five different ASRH core competency guidelines. Each guideline varies in terms of the presentation and arrangement of the competencies by domains, thereby creating numerous core competency items that included thorough individual requirements for the measurement of competency skills. A total of 54 items were chosen and pooled according to the domains identified from the local expert consensus and agreement based on the Ministry of Health of Malaysia ASRH guidelines [23]. Each item was designed to measure knowledge, opinions, and attitudes in managing ASRH using a Likert scale from 0 (strongly not confident) to 4 (strongly confident). The items of each domain were grouped to encourage respondents to focus and allow them to select the most appropriate answer to indicate their positive strength of agreement or feeling of connection to the services delivered. All items were in positive phrase order to avoid confusion. Grouping item responses by adding them demonstrated more reliable measurement. Scores range from 0 to 100, with a score of 0 indicating an individual was not competent and 100 indicating competence in ASRH. Higher scores indicated better knowledge, attitudes, and opinions on the four domains: Domain 1: self-perceived ability to give ASRH education (HE = 13 items); Domain 2: self-perceived capability in ASRH management (C = 11 items); Domain 3: self-perceived adequate ASRH knowledge in decision-making (K = 11 items), and Domain 4: self-perceived appropriate ASRH management attitude (A = 5 items).

Currently, many SRH courses are available online as free training or with a minimum course fee to enroll. Most of the courses are provided through universities in non-Muslim countries. Malaysia is very ethnically diverse, with Islam being the main religion. Teaching ASRH is emphasized in Islamic practice. However, it is not implemented well at school, as it is considered a sensitive subject and teachers are unprepared to teach the SRH topic due to insufficient knowledge [28]. The religious scholars who normally teach pupils attended Islamic schools, did not receive any formal structured SRH training for children. Meanwhile, parents teach SRH to children in a limited scope based on convenience, life experience, and understanding. The poor shared knowledge of SRH risk prevention among adolescents will result in health problems because of not being alert to the risk of danger and the tendency to delay seeking medical or health assistance. Tailoring ASRH education based on local needs, as reported in public health evidence, is highly important. The HCPs at PHCs are at the forefront of health screening. Therefore, to overcome the issue of inadequate ASRH education, it is essential that professional PHC training be assessed based on a validated tool that can provide a structured framework for addressing ASRH needs.

All of the domains in the questionnaire demonstrated Cronbach’s alphas > 0.9, which is considered high and may suggest high redundancy, whereas some items might test the same question under different wording. The high Cronbach’s alpha demonstrates that the test should be shortened. The evidence revealed that the acceptable alpha values should fall between 0.70 and 0.95, with a recommended maximum alpha value of 0.90 [29]. However, we opted to retain the remaining items due to the need to measure different subjects relevant to ASRH core competencies. Removing some items may cause the tool to lose its ability to assess fundamental competencies, as many ASRH services and problems are involved. The assessment tool should measure the essential competencies needed to provide quality SRH services to adolescent clients at PHCs, especially in decision-making and planning.

The high number of items in the tool initially raised concerns among the participants due to the amount of time needed to complete the assessment. However, the face validity assessment showed that the ASRH_CAT is user-friendly, self-administered, and self-explanatory and can be completed within less than 10 min. It also can be incorporated with other study variables and parameters, such as sociodemographic variables, working and training experiences, and practice in ASRH service, to assess associations with the level of competency. 

The questionnaire was constructed and followed the content validation stages [21]. It focuses on decision-making and management planning while recognizing the core competencies of ASRH service in PHCs and the factors that may affect competencies, such as training experience, work experience, education level, and job title. From this study, some essential factors regarding the unanimity of the instrument scores tested within the target population need to be inferred. First, the high level of agreement may be because the respondents recognized that ASRH features essential competencies common to HCPs at PHCs across geographic locations, such as a national and international standardized approach for developing and implementing competencies [30]. Second, the sample mainly included nurses (77%) who had an average of 12.5 years of experience working at PHCs. Only 7% of the respondents had been assigned to adolescent health clinics, and approximately 10% had received at least one formal ASRH training. They were more likely to report moderate to high competencies for the ASRH service, as they may learn the skills through observation and informal training [4].

It is essential to acknowledge that no single assessment model can evaluate all competencies and that different models may measure similar competencies differently and with different levels of precision given their measurement properties. Ideally, the optimum plan will identify multiple assessment models by combining low- and high-fidelity approaches relevant to the competencies to be measured and considering the different stages in professional development and practice. Where possible, care should be taken to avoid one-shot testing, through reliance on a single assessment model, in making critical decisions at any stage of professional development [31].

This tool has its limitations, in that it may not fully cover all the competencies and skills warranted for the HCPs at PHCs based on WHO guidelines, as such assessment tools may be lengthy and time-consuming. Nevertheless, this tool is sufficient for assessing the essential ASRH competencies in decision-making and planning in Malaysian PHC settings for self-competency monitoring. This tool is not meant to be used as a single-measurement tool to assess competency. Repeated assessment may be needed after some exposure at work, following training, or even annually, as deemed appropriate by the organization [32]. This study’s limitation is that this is the first version of the tool validated among the HCPs working in the PHCs serving the Kedah state population alone, and most participants were nurses. Considering the current situation at PHCs during the COVID-19 pandemic, which has caused HCPs to work overtime and burn out from the heavy workload, such feedback from the participants is valuable.

Ground-level implementation is anticipated for this tool to be used to assist HCPs and administrations in choosing HCPs for training or courses. The advantage of this tool is that it can be implemented as a new approach to assess the HCP’s competency before and after the training instead of using the participant’s satisfaction rating of the training itself, as previously conducted. Additionally, this tool was developed based on the targeted population at the primary healthcare level and contains items appropriately validated according to the training modules for assessing the ASRH competency of HCPs at PHCs. Even though the validation process was conducted during the COVID-19 pandemic, the HCPs are capable of performing their work according to their competency requirements [33]. The ASRH-CAT is planned to be used as a baseline assessment of any adolescent health program, to monitor ASRH staff competency levels, and to plan training courses. Future studies with larger samples from the general population are recommended to test its psychometric properties to improve reliability tests and compare with our tool (Appendix A). Other countries can use the questionnaire and validate it in their population after it is translated into their own language.

## 5. Conclusions

The ASRH Competency Assessment tool (ASRH_CAT) was designed to identify ASRH competencies among HCPs in Malaysian PHCs and exhibited satisfactory content validity. This study presents a significant contribution by allowing for the measurement of the competencies through a self-administered instrument. It can be used as the first step in identifying strengths and gaps in knowledge, opinions, and attitudes of ASRH care, thus aiding in future strategic planning for care quality and training plans.

## Figures and Tables

**Table 1 healthcare-11-01116-t001:** CVI assessment for tool items based on consistency (CON), representativeness (REPRESENT), relevance (REL), and clarity (CLA), by domain.

	Code	Construct	I-CVI Score				Mean (±SD)
			CON	REPRESENT	REL	CLA	
Domain 1: Self-perceived ability in ASRH health education Objective: to measure own perception of the ability to provide health education to adolescent clients Statement: I can provide health education to the adolescent client regarding the scope as listed:							
1	HE1	ASRH service access.	1.00	1.00	1.00	0.80	3.44 (0.899)
2	HE2	Availability of Sexually Transmitted Diseases (STI) screening test.	1.00	1.00	1.00	1.00	3.57 (0.840)
3	HE3	HIV screening test service.	1.00	1.00	1.00	1.00	3.64 (0.796)
4	HE4	Information on abortion	1.00	0.80	0.80	1.00	2.87 (1.269)
5	HE5	Contraception methods, including emergency contraceptive measures for adolescents.	1.00	1.00	1.00	0.80	3.42 (1.011)
6	HE6	Guide on proper condom usage.	0.80	0.80	0.80	0.80	3.53 (1.078)
7	HE7	STI prevention measures.	1.00	0.80	0.80	1.00	3.76 (0.909)
8	HE8	HIV prevention measures.	1.00	0.80	0.80	1.00	3.73 (0.875)
9	HE9	Reproductive system development in adolescents.	1.00	1.00	1.00	1.00	3.75 (0.918)
10	HE10	Adolescent services including procedures and management at the clinic	1.00	1.00	1.00	1.00	3.62 (0.888)
11	HE11	Information on risk sexual activities prevention and consequences	1.00	1.00	1.00	1.00	3.66 (0.910)
12	HE12	Gender identity and sexual orientation (including lesbian, gay, bisexual, and transgender)	1.00	1.00	1.00	1.00	3.33 (0.865)
13	HE13	safe sex practice	1.00	1.00	1.00	1.00	3.66 (0.985)
Domain 2: Self-perceived capability in ASRH management Objective: To measure self-perceived capability in ASRH managing and decision-making skills according to guideline practice standards for PHC Statement: I am capable of to…							
14	C1	take SRH history from an adolescent by exploring possible undisclosed issues.	1.00	1.00	1.00	1.00	3.30 (0.889)
15	C2 *	perform a psychosocial assessment to detect risk factors in adolecents’ social, educational, and home environments.	0.80	0.80	0.80	0.40	3.21 (0.941)
16	C3	explain the procedure involved in adolescent health services to the client	1.00	1.00	1.00	1.00	3.38 (0.925)
17	C4	conduct a physical examination when needed for adolescent growth and development assessment	1.00	1.00	1.00	0.80	3.33 (0.944)
18	C5 *	advocate the rights of individuals during clinical examination, including keeping the client informed, consenting to specimen collection, and taking steps to minimize discomfort.	1.00	0.80	0.80	0.40	3.45 (0.923)
19	C6 *	advocate adolescents with special needs and their parent/guardian on adolescent services and management.	0.80	0.60	0.80	0.40	3.46 (0.872)
20	C7	provide a trustful consultation with rights to privacy and confidentiality	1.00	1.00	1.00	0.80	3.69 (0.899)
21	C8	seek consultation before any referral made	1.00	1.00	1.00	1.00	3.67 (0.889)
22	C9	adhere to local policy and guidelines of ASRH.	1.00	1.00	1.00	1.00	3.70 (0.884)
23	C10	provide pregnancy care for adolescents if needed.	1.00	1.00	1.00	1.00	3.70 (0.928)
24	C11 *	provide postnatal care for adolescents if needed.	0.80	0.60	0.60	0.60	3.70 (0.928)
25	C12	provide contraceptive services, including emergency contraceptive measures to adolescents if needed.	1.00	1.00	1.00	1.00	3.55 (0.992)
26	C13	provide a treatment plan for STIs in adolescents if needed.	1.00	1.00	1.00	1.00	3.33 (0.970)
27	C14 *	summaries the main points at the end of the consultation session on risks identified, treatment/management/procedure needed, and decision support for counseling.	0.80	0.80	0.60	0.60	3.40 (0.963)
28	C15 *	request for a chaperone when needed.	0.60	1.00	0.60	1.00	3.49 (0.998)
29	C16 *	support the parents/guardians in their educational tasks (e.g., promotion of a healthy lifestyle, developing adolescent autonomy in following treatment regimens and self-management).	0.80	0.40	0.60	0.60	3.66 (0.906)
30	C17	I can communicate with all the stakeholders about the value of providing respectful, confidential health services to adolescents.	1.00	1.00	1.00	0.80	3.59 (0.924)
31	C18 *	work effectively with schools and other community-based services caring for adolescents in a structured approach for follow-up and referral.	0.40	1.00	1.00	0.40	3.61 (0.894)
32	C19 *	adapt care needed based on the socio-economic and cultural conditions while protecting adolescents’ rights.	0.80	0.60	0.80	0.40	3.56 (0.894)
33	C20	provide appropriate care for a sexually abused adolescent.	1.00	1.00	1.00	0.80	3.43 (0.944)
Domain 3: Self-perceived adequate ASRH knowledge in decision making Objective: to measure self-perceived knowledge required for ASRH Statement: I know how to…							
34	K1	determine appropriate diagnostic tests, including for sexual assault cases.	1.00	1.00	1.00	1.00	3.25 (0.969)
35	K2	diagnose pregnancy in adolescents, including interpreting pregnancy tests.	1.00	1.00	1.00	1.00	3.46 (1.086)
36	K3	diagnose STI in adolescent	1.00	1.00	1.00	1.00	3.22 (1.073)
37	K4	perform appropriate timing to do clinical procedures	0.80	0.80	1.00	1.00	3.49 (0.951)
38	K5	prepare the adolescent client for examination procedures	1.00	1.00	1.00	1.00	3.52 (0.968)
39	K6	use the standards protocols underpinning care in ASRH.	1.00	1.00	1.00	1.00	3.47 (0.891)
40	K7	advice usage of various contraceptive methods	1.00	1.00	1.00	1.00	3.70 (0.825)
41	K8	address related factors that influence ASRH care delivery decision-making (e.g., age, gender, policies).	1.00	1.00	1.00	1.00	3.44 (0.935)
42	K9	Apply the ASRH-related laws in Malaysia i.e., Child Act (2001) and Child Act Amendment (2016)	1.00	1.00	1.00	1.00	3.10 (0.974)
43	K10	Apply the local policies for the provision of ASRH service.	1.00	1.00	1.00	1.00	3.27 (0.962)
44	K11	Assess for ASRH risk factors	1.00	1.00	1.00	1.00	3.44 (0.966)
Domain 4: Self-perceived appropriate ASRH management attitude Objective: To measure self-perceived attitudes toward ASRH clients Statement: I am …							
45	A1 *	comfortable managing adolescent diversity (a certain age group, unmarried, HIV positive, or adolescent with STI).	1.00	1.00	1.00	0.40	3.20 (1.006)
46	A2 *	confident in providing health services for adolescents with SRH problems	0.80	0.80	0.60	0.60	3.46 (0.932)
47	A3	not judge adolescents with societal SRH problems such as pregnancy out of wedlock, HIV positive status, having STI, practicing different sexual orientations, or are gender dysphoria	1.00	1.00	1.00	1.00	3.70 (0.869)
48	A4	ready to provide appropriate treatment services to all adolescent clients with SRH problems regardless of their background	1.00	1.00	1.00	1.00	3.85 (0.937)
49	A5 *	ready to give the best services to adolescent clients with HIV positive or AIDS.	1.00	0.40	1.00	0.40	3.90 (0.850)
50	A6	Able to communicate SRH issues with adolescents.	1.00	1.00	1.00	0.80	3.70 (0.933)
51	A7 *	ready to counsel or prescribe contraceptive methods for adolescents who are sexually active regardless of their marital status.	1.00	1.00	1.00	0.40	3.56 (1.025)
52	A8	able to give non-judgmental counseling to an adolescent who has been sexually abused.	1.00	1.00	1.00	0.80	3.65 (0.974)
53	A9 *	confident to provide comfort when discussing topics related to adolescent sexuality (e.g., sexual orientation, sexual development)	1.00	0.40	1.00	0.40	3.64 (0.932)
54	A10	not take the adolescent’s SRH issues personally.	1.00	1.00	1.00	1.00	3.74 (0.973)

* item was removed based on an I-CVI value < 0.8.

**Table 2 healthcare-11-01116-t002:** Characteristics of the HCPs respondents.

Variables	N = 240	Percentage (%)
**Mean age (standard deviation) year**	38.37 (s.d 6.51)	
**Age category (years)**		
21–30	22	9.17
31–40	136	56.67
41–50	72	30.00
51–60	10	4.16
**Gender**		
female	207	86.2
male	33	13.8
**Ethnicity**		
Chinese	5	2.1
Indian	8	3.3
Malay	224	93.3
Others (Bumiputera)	3	1.3
**Religion**		
Buddhist	4	1.7
Christian	4	1.7
Hindu	7	2.9
Islam	225	93.8
**Highest education level**		
Upper secondary school (SPM)	14	5.8
Diploma level (certificate, STPM)	165	68.8
Degree level (first degree, master, and PhD)	61	25.4
**Occupation**		
Nurses	176	73.3
Doctor	51	21.3
Medical assistant	13	5.4
**Length of health service (year)**		
<5	34	14.2
6–10	71	29.6
11–15	60	25
16–20	40	16.6
21–25	29	12.1
26–30	6	2.5
Mean length of services (SD) year	12.46 (6.50)	

**Table 3 healthcare-11-01116-t003:** EFA and reliability analysis, ITC, and domain-specific Cronbach’s α.

Item Code/ Domain	EFA				ITC	Cα
	Factor Loading into External Factor					
	1	2	3	4		
**Domain 1: Self-perceived ability to give ASRH education**						
HE1			0.925		1.000	0.932
HE2			0.631		0.700	
HE3			0.529		0.488	
HE4			0.516		0.468	
HE5			0.710		0.462	
HE6			0.815		0.487	
HE7			0.654		0.564	
HE8			0.391		0.321	
HE9			0.650		0.517	
HE10			0.625		0.577	
HE11			0.544		0.569	
**HE12**			**0.392**		**0.356**	
HE13			0.712		0.557	
**Domain 2: Self-perceived capability in ASRH management**						
C1	0.629				1.000	0.949
C3	0.668				0.694	
C4	0.737				0.643	
C7	0.636				0.600	
C8	0.572				0.593	
C9	0.630				0.597	
C10	0.563				0.541	
C12	0.512				0.512	
C13	0.492				0.594	
C17	0.696				0.622	
C20	0.754				0.607	
**Domain 3: Self-perceived adequate ASRH knowledge in decision-making**						
K1		0.743			1.000	0.946
K2		0.722			0.659	
K3		0.810			0.676	
K4		0.705			0.658	
K5		0.700			0.568	
K6		0.584			0.573	
K7		0.422			0.434	
K8		0.642			0.633	
K9		0.628			0.626	
K10		0.599			0.619	
K11		0.591			0.581	
**Domain 4: Self-perceived appropriate ASRH management attitude**						
A3				0.697	0.513	0.905
A4				0.760	0.479	
A6				0.682	0.624	
A8				0.589	0.575	
A10				0.653	0.587	

Abbreviations: EFA = exploratory factor analysis; ITC = item-total correlation; Cα = Cronbach’s alpha.

**Table 4 healthcare-11-01116-t004:** Scores of item validity.

	Item Difficulty Index (*p*-Value: Proportion Got It Right)	Item Discrimination Index (Differentiate between Competent and Non-Competent Respondents)
Domain 1: Self-perceived ability to give ASRH education (13 items)	0.7	0.6
Domain 2: Self-perceived capability in ASRH management (11 items)	0.4	0.4
Domain 3: Self-perceived adequate ASRH knowledge in decision-making (11 items)	0.7	0.5
Domain 4: Self-perceived appropriate ASRH management attitude (5 items)	0.6	0.5

## Data Availability

The data presented in this study are available on request from the corresponding author.

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
