# Peer review of "Development and Validation of the Adolescent Sexual and Reproductive Competency Assessment Tool (ASRH-CAT) for Healthcare Providers"

_healthcare, 2023, doi:10.3390/healthcare11081116_

Round 1
Reviewer 1 Report
A very interesting and necessary project with a professional approach to questionnaire development, combining quantitative and qualitative analysis. Another advantage is the link to the WHO guidelines on core competencies for SRH in primary care. It is worth noting in the introduction whether this is, considering the knowledge available to the authors, the first such tool, or whether there are others in use around the world.
However, despite my positive opinion. a few points need to be clarified and described in more detail.
MAJOR CONCERNS
Please specify which phases of the project the description refers to. I feel that the results of step 8 (partially) and step 9 (completely) are missing at this stage. There are no regression analyses listed as step 9. There is no description of what is meant by "future outcomes" in line 167. So, it would be worthwhile to introduce a subsection in the discussion about planned next steps. If regression analyses or correlation with other tools (outcomes) are to be conducted, this could be included in further plans. Similarly, there is no description of suggested scoring methods or plans to conduct CFA, which requires a separate survey. It is unclear whether one overall index or 4 sub-indices are suggested.
The psychometric analysis of the items did not include the distribution of responses to individual questions, including the ceiling and floor effect. Another point to be included in further plans along with preliminary results regarding the indexes obtained.
The phases and steps of the project are precisely listed. The implementation period probably refers only to the main survey. Please also provide a general development period for the entire project, including expert consultations.
The level of implementation of step 8 is unclear. Cronbach's coefficients were calculated, but there is also mention of repeatability of results. Was a test-retest conducted to check the consistency of responses over time. The notation "Establishing if responses are consistent when repeated" should be clarified or removed.
Missing from the discussion (perhaps to be included in the section on limitations) was a comment on the possibilities of implementing this tool and other countries. Some questions will need to be revised (like K9), but a tool built in Malaysia could be the basis for broader implementation.
MINOR AND EDITORIAL
The abstract emphasized the process of item elimination. More important information about the final structure of the 38-item scale was missing. The sentence from line 126 fits very well in the abstract.
In Table 3, it can be guessed that the questions marked in bold HE8 and HE12 are the ones removed, suggests marking them with an asterisk and explain in a footnote.
In addition to the wording of the questions, the categories of answers are important, as explained in line 146. I suggest to provide wording of all five categories or at least the the neutral one (can be hard to say, neither yes nor no, etc.).
Bold in line 172 is not necessary.
Line 144. Perhaps better to change item pool into validation survey.
Author Response
our responses are listed in a table for easy reference. we highlighted in yellow the part in which correction has been made.
|
|
Reviewer 1 comment |
Action taken/comment by writer |
Line Page number |
|
1 |
A very interesting and necessary project with a professional approach to questionnaire development, combining quantitative and qualitative analysis.
|
We follow the recommended steps in tool validation which combine the qualitative part (review of literature and experts’ opinion) and quantitative part (reliability and validity assessment). |
115-246 |
|
2 |
Another advantage is the link to the WHO guidelines on core competencies for SRH in primary care. It is worth noting in the introduction whether this is, considering the knowledge available to the authors, the first such tool, or whether there are others in use around the world. |
There is recent evidence on assessing competency levels among HCP at primary care on SRH right. However, to the best of our knowledge, there are no validated tools for assessing the HCP competency in managing the SRH services and making decisions according to guidelines. |
95-111 |
|
|
Please specify which phases of the project the description refers to. The phases and steps of the project are precisely listed. The implementation period probably refers only to the main survey. Please also provide a general development period for the entire project, including expert consultations.
|
We edited the presentation of the result according to the steps mentioned in the methods section. We mentioned the tool development and validation study period under the section methodology.
|
295-322
117 |
|
|
I feel that the results of step 8 (partially) and step 9 (completely) are missing at this stage. There are no regression analyses listed as step 9.
The level of implementation of step 8 is unclear. Cronbach's coefficients were calculated, but there is also mention of repeatability of results. Was a test-retest conducted to check the consistency of responses over time. The notation "Establishing if responses are consistent when repeated" should be clarified or removed.
|
Results of missing steps added.
We clarified the sentences and added missing information on the test-retest.
|
295-322 |
|
|
There is no description of what is meant by "future outcomes" in line 167.
|
We removed the words on future outcomes and added a new description |
222-238 |
|
|
So, it would be worthwhile to introduce a subsection in the discussion about planned next steps. If regression analyses or correlation with other tools (outcomes) are to be conducted, this could be included in further plans.
Another point to be included in further plans along with preliminary results regarding the indexes obtained. Missing from the discussion (perhaps to be included in the section on limitations) was a comment on the possibilities of implementing this tool and other countries. Some questions will need to be revised (like K9), but a tool built in Malaysia could be the basis for broader implementation. |
We added subheadings under discussion and elaborate on future research plan continuity after tool validation.
Study limitation added |
235-237 382-386 429-435
414-419 |
|
|
Similarly, there is no description of suggested scoring methods or plans to conduct CFA, which requires a separate survey. It is unclear whether one overall index or 4 sub-indices are suggested. The psychometric analysis of the items did not include the distribution of responses to individual questions, including the ceiling and floor effect. |
This part was added to the future research plan as this is the first tool to be used in our country and we plan to focus on highlighting the development and its validation at the pilot study area. |
432-435 |
|
|
The abstract emphasized the process of item elimination. More important information about the final structure of the 38-item scale was missing. The sentence from line 126 fits very well in the abstract. In Table 3, it can be guessed that the questions marked in bold HE8 and HE12 are the ones removed, suggests marking them with an asterisk and explain in a footnote.
In addition to the wording of the questions, the categories of answers are important, as explained in line 146. I suggest to provide wording of all five categories or at least the the neutral one (can be hard to say, neither yes nor no, etc.). Bold in line 172 is not necessary. Line 144. Perhaps better to change item pool into validation survey |
We did the correction as suggested
We added * for item deletion
We define it in our final questionnaire (Appendix 1)
Correction made |
25-26
268
339-352 434 |

Reviewer 2 Report
You need to update information about the relation between "poor knoledge and sexual practice". Since the 70´s many research have shown all the opossite that you afirm. Generally girls have more information and less practice. Boys have more practice and less information. You need to compare your afirmations with some more countries.It is no correct to speak still of illness intead of infections.
Materials and Methods are confusing. Even more Discussion and Conclusions. You cannot add information of analysis in the Discussion. The genreal idea of the article is a disorder.
Author Response
we try to respond to your comments and we listed them in the table attached. We yellow-highlighted changes made in the manuscript
|
|
Reviewer 2 comments |
Action taken/comment by writer |
Line |
|
1 |
Results of steps 8 and step 9 completely missing |
Explanation added in steps 8 and 9 in the methods and results section.
|
217-238 295 - 322 |
|
2 |
The psychometric analysis of the items did not include the distribution of responses to individual questions, including the ceiling and floor effect. |
For the present manuscript, we only presented the steps of questionnaire validation using one study site for the sample population. We planned to confirm the psychometric analysis in the second survey (general population survey later) and we mentioned it in the corrected manuscript. |
179-183 185-193 432-435 |
|
3 |
Study period |
Conducted over 12 weeks from June 2021 to August 2021 |
118 |
|
4 |
Missing from discussion – comment on possibilities of implementing this tool and other countries. |
We added to the discussion. |
430-435 |
|
5. |
Notation “establishing if responses are consistent when repeated” should be clarified or removed |
Notation removed |
|

Reviewer 3 Report
This study aimed to validate a competency assessment tool for adolescent sexual and reproductive health (ASRH) services for healthcare providers (HCP) at primary healthcare (PHC) facilities. Overall, this manuscript was well-written and informative. The assessment tool developed in this project also has practical value. I thank the authors for their hard work. I have several questions and comments, which I elaborate on below.
1. The authors argued that “Using self-assessment evaluation for assessing HCP competency will encourage staff to improve their performance” (lines 79-80). However, I think the authors should have provided a more detailed discussion about why and how using self-assessment evaluation is important. The authors may provide more/stronger justification for developing the tool.
2. The majority of participants were nurses. I wonder if the assessment tool should be developed for different healthcare professionals. In other words, is it appropriate to use the same survey across all healthcare professionals? Maybe the authors could run a post hoc analysis to see if the scores differ as a function of participants’ job categories?
3. I wonder how the cultural and societal contexts in which this study was conducted may affect the study results. I suggest the authors provide a brief discussion about the consideration of culture in their scale development and validation.
4. Did the authors intend to use the assessment tool outside Malaysia? Either, I think the authors should highlight the considerations and limitations of culture in this study.
Author Response
we try to respond to all comments listed in the table attached. We corrected the manuscript and highlighted it in yellow for easy reference
|
|
reviewer 3 comments |
|
|
|
1. |
The authors argued that “Using self-assessment evaluation for assessing HCP competency will encourage staff to improve their performance” (lines 79-80). However, I think the authors should have provided a more detailed discussion about why and how using self-assessment evaluation is important. The authors may provide more/stronger justification for developing the tool |
We added more justification and references regarding the importance of self-assessment evaluation. |
40-49 |
|
2. |
The majority of participants were nurses. I wonder if the assessment tool should be developed for different healthcare professionals. In other words, is it appropriate to use the same survey across all healthcare professionals? Maybe the authors could run a post hoc analysis to see if the scores differ as a function of participants’ job categories. |
Most of the primary healthcare staff who deliver the services for adolescents are nurses and the team are led/supervised by a medical doctor. Therefore, most of the respondents in this study were nurses. |
414-418 |
|
3. |
I wonder how the cultural and societal contexts in which this study was conducted may affect the study results. I suggest the authors provide a brief discussion about the consideration of culture in their scale development and validation. |
We added in the manuscript the importance of socio-cultural background and support with evidence. |
36-37
40-49 100-104 357-359 |
|
4. |
Did the authors intend to use the assessment tool outside Malaysia? Either, I think the authors should highlight the considerations and limitations of culture in this study |
The tool can be used outside Malaysia as it is based on WHO guidelines. The questionnaire was validated in the Malaysian population. In future studies, it can be validated in other countries and translated in other language |
373-374 |
|
|
Distribution of responses to individual questions. |
We added an explanation |
179-183 |

Round 2
Reviewer 1 Report
Thank you for considering the comments
Reviewer 2 Report
I accept their explanations and consider now that it is a good article